# Comparative Proteomics and Genome-Wide Druggability Analyses Prioritized Promising Therapeutic Targets against Drug-Resistant *Leishmania tropica*

**DOI:** 10.3390/microorganisms11010228

**Published:** 2023-01-16

**Authors:** Sara Aiman, A. Khuzaim Alzahrani, Fawad Ali, Mohd. Imran, Mehnaz Kamal, Muhammad Usman, Hamdy Khamees Thabet, Chunhua Li, Asifullah Khan

**Affiliations:** 1Faculty of Environmental and Life Sciences, Beijing University of Technology, Beijing 100124, China; 2Medical Laboratory Technology, Faculty of Applied Medical Sciences, Northern Border University, Arar 91431, Saudi Arabia; 3Department of Biochemistry, Abdul Wali Khan University Mardan, Mardan 23200, Pakistan; 4Department of Pharmaceutical Chemistry, Faculty of Pharmacy, Northern Border University, Rafha 91911, Saudi Arabia; 5Department of Pharmaceutical Chemistry, College of Pharmacy, Prince Sattam Bin Abdulaziz University, Al-Kharj 11942, Saudi Arabia; 6Chemistry Department, Faculty of Arts and Science, Northern Border University, Rafha 91911, Saudi Arabia

**Keywords:** leishmaniasis, subtractive proteome analysis, drug targets identification, virtual screening, ADME analysis

## Abstract

*Leishmania tropica* is a tropical parasite causing cutaneous leishmaniasis (CL) in humans. Leishmaniasis is a serious public health threat, affecting an estimated 350 million people in 98 countries. The global rise in antileishmanial drug resistance has triggered the need to explore novel therapeutic strategies against this parasite. In the present study, we utilized the recently available multidrug resistant *L. tropica* strain proteome data repository to identify alternative therapeutic drug targets based on comparative subtractive proteomic and druggability analyses. Additionally, small drug-like compounds were scanned against novel targets based on virtual screening and ADME profiling. The analysis unveiled 496 essential cellular proteins of *L. tropica* that were nonhomologous to the human proteome set. The druggability analyses prioritized nine parasite-specific druggable proteins essential for the parasite’s basic cellular survival, growth, and virulence. These prioritized proteins were identified to have appropriate binding pockets to anchor small drug-like compounds. Among these, UDPase and PCNA were prioritized as the top-ranked druggable proteins. The pharmacophore-based virtual screening and ADME profiling predicted MolPort-000-730-162 and MolPort-020-232-354 as the top hit drug-like compounds from the Pharmit resource to inhibit *L. tropica* UDPase and PCNA, respectively. The alternative drug targets and drug-like molecules predicted in the current study lay the groundwork for developing novel antileishmanial *therapies*.

## 1. Introduction

Leishmaniasis is a neglected tropical disease caused by *leishmania* species, i.e., a sandfly vector-borne parasitic kinetoplastid protozoa [1]. The three major pathological forms of leishmaniasis are kala-azar or visceral leishmaniasis (VL), cutaneous leishmaniasis (CL), and mucocutaneous leishmaniasis (MCL). The most lethal one is VL, characterized by severe inflammatory reactions in the spleen and liver, which might be fatal. CL is the most frequently prevalent that infects the epidermal layer of the skin and leads to disfiguring lesions [2]. Leishmaniasis poses a threat to an estimated 350 million individuals in 98 countries. Globally, 12 million cases are reported annually, with an average of 2 to 2.5 million new infections. Among these, half a million cases are of VL, and 1 to 1.5 million are of CL [3].

*Leishmania tropica* is a flagellated parasite that causes cutaneous leishmaniasis [4]. *L. tropica* is a highly diverse species complex with a wide range of biochemical, serological, and genomic features. The species holds a wide geographical distribution from Africa to Eurasia [5]. According to a survey released in 2018, over 85% of new CL cases occurred in different parts of the world, including Algeria, Afghanistan, Brazil, Bolivia, Colombia, Iraq, Iran, Syria, Pakistan, and Tunisia [3]. The major risk factors for leishmaniasis include inadequate hygiene, poverty, population mobility, malnutrition, and an immunocompromised state [6]. CL causes skin lesions that appear weeks to months after a female sandfly bite, leaving permanent scars behind and leading to serious disabilities.

The clinical symptoms of leishmaniasis vary with respect to the severity of skin lesions and range from simple CL to extreme VL conditions. Some patients have been reported to develop post-kala-azar leishmaniasis after therapy, while nasopharyngeal mucocutaneous leishmaniasis (MCL) rarely occurs [5,6]. Antiparasitic therapeutic options are limited due to drug toxicity, high cost, and increased drug resistance. Unfortunately, there is limited information about drug resistance in *Leishmania*. The treatment of CL is based on chemotherapy with pentavalent antimonials as the predominantly used medications. However, improper doses and treatment have contributed to the emergence of resistance to this agent [7,8]. Sodium stibogluconate (SSG) has been used to treat CL worldwide for decades [9]. However, parasitic resistance to SSG and other antimonials has recently been reported worldwide [10]. Miltefosine is a recently designed drug for CL; however, parasitic resistance against this drug has also been reported [11]. New or alternative antileishmanial therapeutic strategies are indispensable due to multidrug resistance, cost, toxicity, and unresponsiveness to the treatment regimens.

Advancements in bioinformatics and computational biology approaches, in combination with the availability of pathogen genome sequences, have greatly assisted in prioritizing novel therapeutic targets against pathogenic organisms. Multiomics data, such as proteomics, metabolomics, and genomics, have proven tremendous in drug discovery processes by significantly lowering the cost and time required for in vivo and experimental screening [12]. Subtractive and comparative genomic approaches screen the entire proteome of the host and pathogen to prioritize pathogen-specific essential proteins holding therapeutic potential [13,14]. In the present study, we analyzed the recently updated *L. tropica* genome resources and prioritized several druggable protein targets. The analyses ultimately identified UDP-glucose pyrophosphorylase and proliferating cell nuclear antigen as promising alternative antileishmanial targets. The screening of drug-like compound repositories identified several lead compounds as potent inhibitors against these new targets.

## 2. Materials and Methods

Host–pathogen comparative proteomics analyses are reliable for identifying potential therapeutic targets against pathogens. A systematic flowchart depicting all the steps followed during this study is shown in Figure 1.

### 2.1. Protein Sequence Retrieval

The protein sequences of mutidrug-resistant *L. tropica* axenic amastigotes were retrieved from the TriTrypDB- kinetoplastid informatics resource in FASTA format [15]. CD-HIT clustering analysis was conducted to eliminate paralogous proteins with sequence similarity criteria of ≥80% [16].

### 2.2. Essential Proteins Identification

The nonparalogous protein sequences were scanned against the eukaryotic database of essential genes (DEG). The DEG database contains experimentally validated essential genes from eukaryotes [17]. The standalone BLASTp program [18] was used to screen DEG against pathogen protein sequences with an E-value threshold of 10^−4^, bit score ≥100, sequence identity ≥35%, and query coverage ≥35%.

### 2.3. Human Host and Gut Nonhomolog Proteins Identification

The protein sequences were then subjected to BLASTp against the human proteome set acquired from the NCBI database [19] to identify host nonhomologous proteins. Protein sequences with a high degree of homology with the human proteome were eliminated, and human nonhomologous pathogen proteins were prioritized. The resultant nonhomologous proteins were scanned against human gut flora proteins [20] to avoid sequences showing homology to the gut microbiota. The threshold criteria for this BLASTp analysis were ≤35% sequence identity, ≤35% query coverage, and an E-value cut off 10^−4^.

### 2.4. DrugBank Database Screening

The updated DrugBank database was screened to identify novel drug targets of *L. tropica*. A threshold of ≤35% sequence identity and query coverage was applied in this analysis [21]. Parasite proteins depicted as nonhomologous to the DrugBank were listed as alternative drug targets based on the set threshold criteria.

### 2.5. Structure Homologs Search

The PDB database was screened to identify structural information of the prioritized druggable proteins. Parasite proteins holding no structural information in PDB were modeled using the Swiss model by template base homology modeling [22]. The modeled 3D structures were then verified for accuracy using ERRAT and RAMPAGE [23].

### 2.6. Druggability Analyses

Several criteria are used to determine potential therapeutic targets, including molecular weight, function, cellular localization, and virulence factors [24]. The shortlisted prioritized proteins from the above analyses were tested for druggability potential. The subcellular localization of these target proteins was determined using PSORTb v.3.0.2 [25] and CELLO2GO V.2.5 [26] web servers. The drug molecule binding pockets of the targets were identified using PockDrug-server [27].

### 2.7. Pharmacophore-Based Virtual Screening

The Pharmit server was utilized for the pharmacophore-based virtual screening of millions of compounds from the built-in databases, including Molprot, ZINC, ChEMBL, and PubChem [28]. The 3D structure of the UDP-glucose pyrophosphorylase protein (PDB ID: 5NZM) and proliferating cell nuclear antigen (PCNA) (PDB ID: 6J0J) proteins were used for virtual screening. A total of nine pharmacophore features were enabled during the screening, including three aromatic and six hydrophobic features. The results were minimized to a significant level based on the RMSD value to shortlist the top-ranked inhibitors from the millions of drug-like compounds. The top ten hit compounds were later docked in the binding pocket of UDP-glucose pyrophosphorylase (LTRL590_180015300) and PCNA (LTRL590_150020700) using the CB-Dock tool to check their binding feasibility [29]. Protein–ligand interactions were visualized using the Discovery Studio Visualizer [29].

### 2.8. ADME Analysis

The absorption, distribution, metabolism, and elimination (ADME) parameters of the top lead compounds were predicted using SwissADME [30]. ADME provides essential information regarding the drug-like capabilities of compounds. The SMILES format of the compounds was used as input to calculate the ADME parameters, physicochemical descriptors, pharmacokinetic features, and the drug-like nature of the lead compounds.

## 3. Results

### 3.1. Essential Proteins Identification

A total of 8462 nonparalogous complete protein sequences of *L. tropica* strain L590 were subjected to BLASTp against the DEG database to identify pathogen-essential proteins. Essential proteins are indispensable for performing key cellular functions and pathogen survival [31]. The analyses identified 1225 pathogen protein homologs to the DEG entries (Appendix A).

### 3.2. Human Host Nonhomologous Proteins

Pathogen-essential proteins are considered potential drug targets. However, these proteins must be nonhomologous to human proteomes and human gut microbiota proteins to avoid the adverse effects of drugs [32]. The nonparalogous pathogen-essential proteins were scanned against the human proteome. The analysis identified 727 pathogen proteins that were nonhomologous to the human host (Appendix A). Additional comparative sequence scanning against the human gut flora proteome set prioritized 496 pathogen proteins that were nonhomologous to the human gut flora proteome (Appendix A).

### 3.3. DrugBank Database Scanning 

The 496 *L. tropica* essential proteins prioritized in the above steps were scanned against the DrugBank database. Pathogen proteins depicting no homologies to the known drug targets in the DrugBank database may be predicted as new or alternative drug targets [33]. DrugBank database scanning identified 19 already reported and 477 as DrugBank nonhomolog targets among the 496 pathogen-prioritized proteins.

### 3.4. Druggability Analysis

DrugBank nonhomologous proteins from *L. tropica* were subjected to subcellular localization analysis. Subcellular localization prediction is a key aspect of druggability analysis, and cytoplasmic proteins are considered ideal drug targets [34,35] (Figure 2). Therefore, the predicted cytoplasmic proteins were scanned against the PDB database to determine the three-dimensional (3D) structural information of the shortlisted pathogen proteins. However, none of the prioritized protein’s 3D structural information was available in PDB; therefore, the shortlisted proteins’ three-dimensional (3D) structures were modeled via homology modeling using their close structural homologs from PDB as a template. Eighteen *L. tropica* proteins were shortlisted for further druggability analysis. The Swiss model predicted the 3D structures of these proteins and was validated by the ERRAT tool with a quality factor score of >85%, representing high-quality protein models. Likewise, the Ramachandran plot predicted >85% to 90% residues of the modeled structures in the plot’s allowed regions and ensured the proteins 3D structure model’s accuracy [36]. Finally, nine proteins were prioritized based on pocket druggability analysis scores of >0.5 (Table 1).

### 3.5. Pharmacophore-Based Virtual Screening

Among the shortlisted drug target candidate proteins, UDP-glucose pyrophosphorylase (UGPase) (LTRL590_180015300) was prioritized for virtual screening based on druggability analysis and close structural homolog availability from PDB. The LtUGPase showed 100% query coverage and 98% sequences similarity with the Leishmania major LmUGP- murrayamine-I complex (PDB ID: 5NZM), available in PDB. So far, no commercial inhibitors or antileishmanial drugs have been reported based on the UDPase target. LmUGPase from L. major was used as a template for the homology modeling of L. tropica UGPase (LtUGPase). The pharmacophore model designed for the 3D structure of LtUGPase showed three aromatics and six hydrophobic features (Figure 3A). The top-10-hit small molecules were selected based on the docking score and RMSD values (Table 2). These 10 compounds were then subjected to molecular docking against LtUGPase to calculate the binding energies. The active site of LmUGP- murrayamine-I complex comprises four key residues: Arg248, Val371, Pro372, and Arg373 [37]. This information was used for structure-based pharmacophore model designing and drug-like compound databases virtual screening. The top-10-hit compounds were subjected to molecular docking analysis to evaluate the binding conformation of the lead compounds within the active site of the receptor molecule. The MolPort-000-730-162 compound showed significant molecular interactions with the conserved residues (370–380) in the C-terminus of LtUGPase and ranked as the top hit (Figure 3B). Additionally, the docking analysis predicted that all the top-screened compounds exhibited substantial molecular interactions with the receptor (Table 2, Appendix A).

The enzyme LtPCNA showed 100% query coverage and 94% sequence similarity with the crystal structure of PCNA from L. donovani (PDB ID: 6J0J). PCNA from L. donovani was used as a template to model the 3D structure of the LtPCNA protein. A structure-based pharmacophore model was designed against the LtPCNA target that exhibited three hydrogen donor and one hydrogen acceptor pharmacophoric features (Figure 4A). Pharmacophore-based virtual screening predicted several druggable compounds against the LtPCNA target from the Pharmit resource. The top 10 hits were prioritized as drug-like compounds based on the docking scores and RMSD values (Table 3). Molecular docking analysis was performed to calculate the binding energies of these 10 compounds with the residues in the active sites of LtPCNA. Among these, MolPort-020-232-354 was identified as the top lead compound, showing significant hydrogen bond interactions with the LYS193, PRO212, GLY223, and ASN224 residues of LtPCNA (Figure 4B). Furthermore, molecular docking analysis anticipated substantial interactions of all the top-screened compounds with LtPCNA (Table 3, Appendix A).

### 3.6. ADME Analysis

The molecular properties of drug-like small molecules are crucial for the effective drug’s design, synthesis, and clinical applications. The four most important pharmacokinetic parameters are absorption, distribution, metabolism, and excretion (ADME). A lead compound must follow the ADME criteria to be a successful drug [38]. Chemical descriptors based on physiological properties and chemical structures were used to calculate the pharmacokinetic properties of the top 10 hit compounds. Multiple physicochemical properties, including molecular weight, hydrogen bonding, hydrophobicity, reactivity, bioavailability, molecular stability, aqueous solubility values (logP and logS), skin permeability coefficient (log kp), gastrointestinal tract absorption (GI), and blood–brain barrier (BBB) were computed to evaluate the efficacy of these compounds (Appendix A).

Lipinski’s rule of five is one of the most effective models for evaluating a suitable drug based on the solubility and permeability of a compound [39]. Among the top 10 lead compounds, C2-C7 and C10 showed drug-like properties based on Lipinski’s rule of five, whereas C1, C8, and C9 were found to violate the rules. The compounds C1, C8, and C9 showed low gastrointestinal absorption, whereas C2-C7 and C10 exhibited high gastrointestinal absorption. All compounds showed low penetrability through the blood–brain barrier (BBB), except compound C5, which was predicted to be able to cross this barrier. All these compounds were substrates for the p-glycoprotein, which plays an important role in pumping xenobiotics and harmful substances back into the gut lumen, focusing on the propensity of these molecules to be potential inhibitors against LtUGPase [40]. None of these compounds showed any ADME toxicity or mutagenicity. However, compounds C1, C8, and C9 require additional chemical modification to fulfill drug-like properties (Appendix A).

The top hit, drug-like compounds against the LtPCNA target are shown in Appendix A. The compounds C1, C2, C4, and C7-C9 demonstrated drug-like properties based on Lipinski’s rule of five, whereas the compounds C3, C5, C6, and C10 violated Lipinski’s rules. Based on the ADME results, the compounds C2, C4, C5, and C7-C9 exhibited higher gastrointestinal absorption, whereas the compounds C1, C3, C6, and C10 were predicted to have low gastrointestinal absorption. All the compounds were predicted to show low penetrability through the blood–brain barrier (BBB). Compounds C1-C4 and C9 were not predicted as p-glycoprotein substrates, whereas the compounds C5-C8 and C10 were predicted as p-glycoprotein substrates. There was no indication of ADME toxicity or mutagenicity for any of these compounds. Compounds C3, C5, C6, and C10 need additional chemical modifications to possibly exhibit more potent drug-like properties.

## 4. Discussion

Leishmaniasis has been listed as one of the most neglected tropical diseases by the World Health Organization (WHO), for which the development of new therapeutic strategies has become indispensable [41]. The present study analyzed the *L. tropica* proteome to identify suitable therapeutic targets against drug-resistant leishmaniasis. We followed a subtractive proteomic analysis approach to identify the *L. tropica* essential and human host nonhomologous proteins. According to the law of centrality and lethality, the functional perturbation of such proteins might be deleterious for the pathogen’s survival [42,43]. These proteins were further shortlisted based on strict druggability criteria.

Among the shortlisted novel drug targets, UDP-glucose pyrophosphorylase (UGPase) (LTRL590_180015300) catalyzes the conversion of -D-glucose-1-phosphate (Glc-1P) and UTP into UDP-glucose (UDP-Glc), an important metabolite in the carbohydrate pathway of all organisms [44]. UDP-Glc is interconverted into UDP-Gal by catalytic reaction of UDP-Glc 4’-epimerase, which is connected to the biosynthesis of nucleotide sugars, confirming the role of this enzyme in galactose salvage, thus essential for parasite growth [45,46]. UGPase is considered a virulence factor, as it is necessary to manufacture cell surface glycoconjugates [47]. *Leishmania* species express a variety of glycoconjugates on their cell surfaces, which are constantly changing throughout the life cycle of these species. These glycoconjugates are associated with the survival and proliferation of parasites in the insect and the mammalian host [48,49]. A combination of gene deletion and protein destabilization techniques reported that reducing the level of UDPase to a minimum results in growth arrest and, ultimately, the cell death of *L. major* [50]. The structural analysis indicated that understanding the catalytic mechanism of UGPases can provide a template for designing species-specific UGPase inhibitors [47]. UGPase has been reported recently as a drug target against *Leishmania* species [51]. However, no potential drug is commercially available based on UGPase inhibition. Therefore, based on pocket druggability analysis and structural validation scores, the *L. tropica* UDPase protein (LtUGPase) was prioritized in the current study for virtual screening to identify drug-like small molecules to possibly inhibit the activity of this protein. These analyses and ADME profiling predicted MolPort-000-730-162 as a top-ranked molecule among the top-10-hit compounds that may inhibit LtUGPase. Experimental and clinical assays are required for further validation of these results.

Proliferative cell nuclear antigen (PCNA) putative protein (LTRL590_150020700) has also been found among the finally shortlisted druggable protein targets. PCNA plays a vital role in the DNA metabolism process in eukaryotes. PCNA augments polymerases’ processivity by providing an *anchor* point for DNA *polymerases* and acting as a DNA sliding clamp protein. PCNA is necessary for DNA repair and recombination as well [52,53]. Studies have shown that PCNA is highly expressed in drug-sensitive and drug-resistant *L. donovani* strains in clinical isolates [54]. Pentavalent antimony is a commonly used drug against *Leishmania* infection. PCNA has a critical role in avoiding DNA damage caused by antimonials and is reported in association with antimony resistance in *Leishmania spps* [55,56]. Several lead inhibitors are predicted in the current study against the *L. tropica* PCNA target. Additional in vivo and in vitro assays are required to validate the antileishmanial efficacy of these predicted lead compounds.

3-mercaptopyruvate sulfurtransferase (LTRL590_050014400), identified as a potential drug target in the current study, is involved in thioredoxin and antioxidant metabolism [57]. This protein is reported to tolerate oxidative stress, i.e., caused by the host’s immune system in *Leishmania* species [58]. It is responsible for the control of reactive oxygen species cytotoxicity in aerobic tissues [59]. The LTRL590 070011300 gene encodes a highly conserved ribosomal protein and is also shortlisted as a drug target in the current study. This ribosomal protein is a part of the 60S subunit of the ribosome responsible for rRNAs and protein domains unique to kinetoplastid ribosomes [60]. Keeping in view the evolutionary distinction between *Leishmania* and humans, the *Leishmanial* ribosome structure may open up new possibilities for developing potent antileishmanial therapies [61].

Tyrosyl-tRNA synthetase putative (TyrRS) (LTRL590_140021400) has also been shortlisted in the current analysis. TyrRS belongs to a family of aminoacyl-tRNA synthetases (aaRSs) involved in essential biological processes, including nucleotide binding, aminoacyl-tRNA ligase activity, tRNA aminoacylation for protein translation, and cellular proliferation [62]. TyrRS has been reported to play a possible immunomodulating role by inducing a proparasitic response and inflammatory phagocyte recruitment during *Leishmania* infections [63]. The E2-like ubiquitin conjugation enzyme (LTRL590_150018200) is also prioritized as a potential drug target that participates in the ubiquitination pathway and regulates various cellular functions [64]. Several ubiquitination pathway proteins have previously been reported as potential drug targets in trypanosomatid diseases [65,66].

Protein tyrosine phosphatase-like protein (PTP) (LTRL590_160007700) has also been shortlisted as a therapeutic target in the current analysis. PTPs have been implicated in controlling various cell activities, including eukaryotic cell proliferation and differentiation [67]. *Leishmania* PTP is involved in metabolic pathways that are involved in amastigote survival, differentiation, and enhancing virulence and disease progression in the host [68,69]. Structural analysis of *Leishmania* PTP shows uniquely remarkable conservation in the active sites of these proteins [70]. This information could be used to develop novel inhibitors to target parasite-specific PTPs.

ADP-ribosylation factor-like ARL protein 1 (LTRL590_170005800) is a tiny cytoplasmic guanosine-5′-triphosphate (GTP)-binding protein that has been associated with a variety of endo- and exocytic vesicular transport processes [71]. ARF undergoes conformational changes when binding with nucleotide, which modulates the affinity of ARL to bind with other proteins, lipids, or membranes [72]. The ARF protein is essential in vesicular trafficking and structural maintenance of the Golgi network in the eukaryotic cell [73]. ARF proteins are highly conserved among the trypanosomatids [74] and are associated with intracellular trafficking [75]. Inhibiting the activity of this protein could halt the pathogenesis of *L. tropica* and assist in developing novel therapeutic reagents.

The ADP/ATP carrier protein 1 (AAC), mitochondrial precursor, putative (LTRL590_190006600) protein is involved with the essential metabolic process of transporting ADP into the mitochondrial matrix and ATP to fuel the cell by maintaining high cytosolic ATP concentrations for energy-requiring metabolic processes [76]. Mitochondrial carrier proteins have been reported as drug targets against *Trypanosoma brucei* and *Saccharomyces cerevisiae* [77]. However, this protein has not been reported before as a promising drug target against *Leishmania* species. The analyses of the current study predict this protein as the promising druggable candidate to combat *L. tropica* infection.

The druggable proteins prioritized in the current study are highly conserved and important for the survival, growth, and virulence of the *Leishmania* parasite within the host. These proteins may therefore implicate potential therapeutic targets in the future. The exploration of refining the 3D structures of these proteins may lay the groundwork for suitable antileishmanial drug development. The in silico approaches used in this study could pave the way to identify novel therapeutic targets and develop species-specific potent drugs that aid in eliminating many parasitic diseases.

## 5. Conclusions

In the present study, we used the entire proteome data of drug-resistant *L. tropica* to determine new and potent therapeutic targets against *L. tropica* infection. A comparative subtractive genome approach was used to identify parasite-specific essential genes involved in the metabolic pathways responsible for pathogen survival, proliferation, and virulence. In silico druggability analysis prioritized several novel drug targets against *L. tropica* that have not been previously reported. Pharmacophore-based virtual screening of updated biological databases, ADME evaluation, and docking studies prioritized several drug-like small compounds against newly identified LtUGPase and LtPCNA targets that may be tested in the future with respect to antileishmaniasis activity.

## Figures and Tables

**Figure 1 microorganisms-11-00228-f001:**
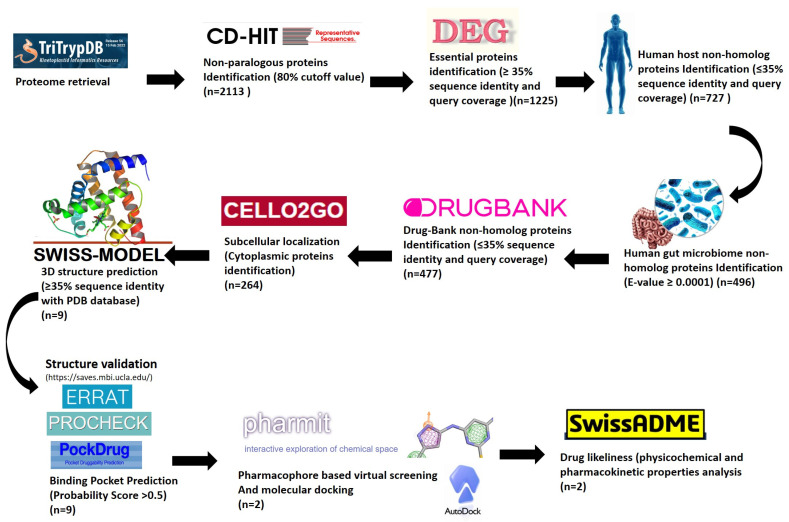
Flowchart diagram of the methodological steps pursued during this study. ‘n’ represents the number of proteins shortlisted in each step.

**Figure 2 microorganisms-11-00228-f002:**
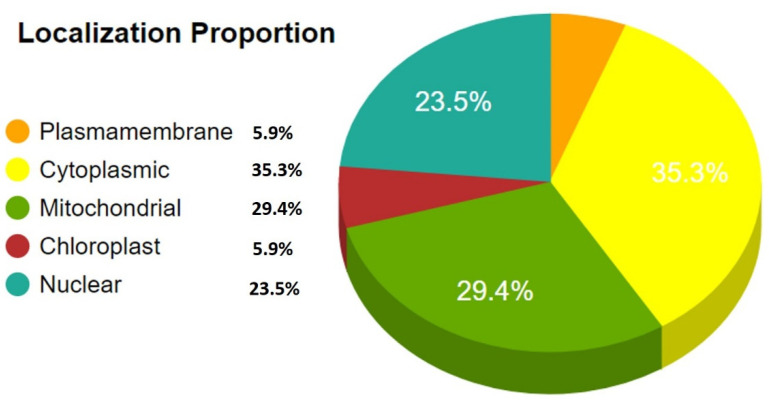
The subcellular localization prediction of the *L. tropica* essential, human nonhomolog proteins.

**Figure 3 microorganisms-11-00228-f003:**
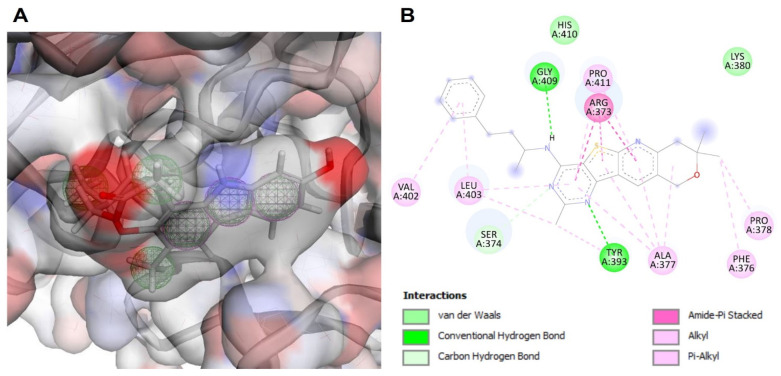
(**A**) The pharmacophore design based on the active site of LtUGPase. (**B**) The molecular interaction of top hit compound docked in the active site of LtUGPase. The nature of protein–ligand interactions is represented with different colors.

**Figure 4 microorganisms-11-00228-f004:**
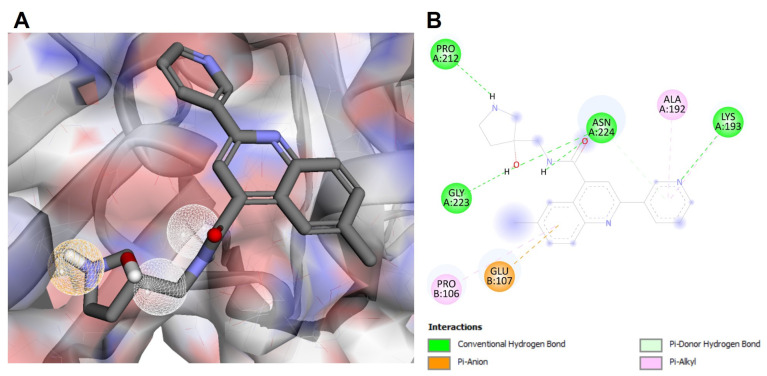
(**A**) The pharmacophore model designed based on the active site of LtPCNA. (**B**) The molecular interaction of top hit compound docked in the active site of LtPCNA. The nature of protein–ligand interactions is represented with different colors.

**Table 1 microorganisms-11-00228-t001:** The molecular modeling, 3D structure validation, and druggability analysis of the prioritized shortlisted proteins of *L. tropica*.

Tritrypdb IDs	Protein Name	PDB HomologousID’s	ERRAT Quality Factor	QMEAN > −4	PockDrug Score > 0.5 (Residues in Pocket)	Molecular Weight (Dalton)	Ramachandran Plot
LTRL590_050014400	3-mercaptopyruvate sulfurtransferase	1okg.1.A	88.4615	−0.69	0.85 (14)	40,141.65	88.30%
LTRL590_070011300	60S ribosomal protein L7a, putative	5t2a.40.A	93.9394	−1.19	0.99 (15)	29,763.01	90.50%
LTRL590_140021400	Tyrosyl-tRNA synthetase, putative	3p0h.1.A	96.5997	−0.85	0.91 (27.0)	74,978.22	92.50%
LTRL590_150018200	E2-like ubiquitin conjugation enzyme, putative	3kpa.2.A	98.6301	−0.08	0.88 (15.0)	19,364.13	95.50%
LTRL590_150020700	Proliferative cell nuclear antigen (PCNA), putative	6joj.2.A	92.956	−2.80	0.59 (25.0)	32,412.73	85.40%
LTRL590_160007700	Protein tyrosine phosphatase-like protein	3s4o.2.A	96.4029	−0.82	0.86 (18.0)	19,513.53	95.40%
LTRL590_170005800	ADP-ribosylation factor-like protein 1	2x77.1.A	96.319	−0.62	0.87 (10.0)	20,820.83	92.60%
LTRL590_180015300	UDP-glucose pyrophosphorylase	5nzm.1.A	92.4612	−0.67	0.97 (17.0)	54,481.27	93.70%
LTRL590_190006600	ADP, ATP carrier protein 1, mitochondrial precursor, putative	6gci.1.A	97.5	−2.14	0.65 (29.0)	35,097.85	96.80%

**Table 2 microorganisms-11-00228-t002:** CB-Dock scores and RMSD values of top 10 hit molecules obtained against UGPase target from virtual screening using Pharmit server.

Serial No.	Compounds (MolPort IDs)	Molecular Weight (g/mol)	RMSD (Å)	Molecular Formula	CB-Dock (Vina Score)
C1	MolPort-002-619-190	456.71	0.322	C25H36N4S2	−6.7
C2	MolPort-000-451-699	444.531	0.324	C27H28N2O4	−7.2
C3	MolPort-000-730-162	432.59	0.331	C25H28N4OS	−8.6
C4	MolPort-000-451-697	446.503	0.334	C26H26N2O5	−7.1
C5	MolPort-000-451-711	476.548	0.335	C28H29FN2O4	−7.2
C6	MolPort-000-451-749	397.427	0.337	C22H23NO6	−7.9
C7	MolPort-002-611-137	339.395	0.338	C19H21N3O3	−8
C8	MolPort-002-619-190	456.71	0.350	C25H36N4S2	−6.9
C9	MolPort-002-608-446	442.68	0.351	C24H34N4S2	−6.7
C10	MolPort-000-451-699	444.531	0.377	C27H28N2O4	−6.9

**Table 3 microorganisms-11-00228-t003:** CB-Dock scores and RMSD values of top 10 hit molecules obtained against LtPNCA target from virtual screening using Pharmit server.

Serial No.	Compounds (MolPort IDs)	Molecular Weight (g/mol)	RMSD (Å)	Pharmit Score	Molecular Formula	CB-Dock (Vina Score)
C1	MolPort-001-741-093	424.402	1.826	−6.01	C20H24O10	−7.2
C2	MolPort-000-700-443	385.85	1.529	−5.92	C20H20ClN3O3	−6.7
C3	MolPort-047-116-128	342.264	1.750	−5.84	C12H14N4O8	−6.7
C4	MolPort-039-345-350	374.345	1.812	−5.84	C19H18O8	−7.5
C5	MolPort-002-525-976	342.297	1.990	−5.83	C12H22O11	−6.8
C6	MolPort-046-836-802	414.36	1.964	−5.79	C15H26O13	−6.4
C7	MolPort-020-232-354	362.433	1.537	−5.74	C21H22N4O2	−7.1
C8	MolPort-020-232-872	313.361	1.442	−5.73	C16H19N5O2	−7.2
C9	MolPort-004-860-220	434.467	1.269	−6.72	C25H23FN2O4	−7.2
C10	MolPort-044-727-363	432.381	1.986	−6.75	C21H20O10	−7.6

## Data Availability

All of the relevant data are provided in the form of regular figures, tables, and Appendix A.

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
