# Peer review of "Comparative Proteomics and Genome-Wide Druggability Analyses Prioritized Promising Therapeutic Targets against Drug-Resistant Leishmania tropica"

_microorganisms, 2023, doi:10.3390/microorganisms11010228_

Round 1

Reviewer 1 Report

The article “Comparative proteomics and Genome-wide druggability analyses prioritized promising therapeutic targets against drug-resistant Leishmania Tropica” takes an in silico approach to analyze proteomic data bases to identify essential cellular proteins in the parasites. An analysis of the proteome data repository for L. tropica, together with a druggability analysis and ADME profiling was used to identify potential targets and inhibitor candidates.

Although the entire work is based on modeling and predictions with no actual proof of concept experiments, the work is still thorough, informative, and useful to the scientific community. The manuscript is well written.

The title of the manuscript states claims that the analysis focuses on “drug-resistant Leishmania Tropica”. This does not seem to be the case. Where in the data analysis flow is a step that selects for proteins present in drug-resistant parasites? The only mention is in the discussion, as one of their identified candidate proteins “is reported in association with antimony resistance in Leishmania spps” (line 306).

Impressive are the thorough steps of using available databases and informatics tool. Figure 1 is a nice visual overview. I suggest to add either to this figure or as a separate figure the number of candidates identified – or narrowed down to – for each step. This is for the most part present in the written result section.

The authors refer to additional file 1 “The analyses identified 1,225 pathogen proteins homologs to DEG entries (Additional File 1).” Where is this file? It is not in the supplemental material.

Figure 2: Please clarify if this is a representation of the 496 candidates identified after the DrugBank screen.

In the result section, the authors state that the selection of UDPase protein (LtUGPase) was based on “was prioritized for virtual screening based on druggability analysis and close structural homologs availability from PDB” (line 191-192). But in the discussion section they state that (one of) the reason for selecting this candidate is because the protein has already been identified as a drug target but that not no potential drug is available. This should also be stated in the result section, if it was indeed a determining factor in selecting UGPase as lead candidate.

Please discuss how available the identified lead compounds are for follow up research and potential commercialization.

Overall the paper is well written. I noticed though that Leishmania is not consistently italicized.

Reviewer 2 Report

The manuscript addresses an organized and clearly explained metodology trying to find new therapeutic targets against leishmania.

I have some questions:

- Have the authors used the proteome of promastigotes? Wouldn't it be better to utilize the proteome of amastigotes? 

- The authors have discarded human gut microbiota proteins, thus, it is expected oral administration of the drug: How many proteins have been discarded with this filter?

- What is the specie specificity of these proteins? Have the authors analysed them in other Leishmania spp?

- UDP glucose pyrophosphorylase have been previously described. It could be interested to perform a virual screening of compounds for any of the other possible identified targets. 

Round 2

Reviewer 2 Report

It is suitable for publication